

# Phylogeography of the rare and endangered lycophyte *Isoetes yunguiensis*

Tao Zheng, Xuanze He, Honghuan Ye, Wei Fu, Maimai Peng and Guangqian Gou

College of Life Sciences, Guizhou University, Guiyang, Guizhou, China

## ABSTRACT

**Background**. *Isoetes yunguiensis* Q. F. Wang & W. C. Taylor is a lycophyte of an ancient genus, and it is endemic to China. It is a first-class protected plant in China. This living fossil is used in paleoecology and studies on the evolution of Lycophytes in the Yunnan-Guizhou Plateau. In recent years, human activities have caused the disappearance of several wild populations, and the number of plants in the existing populations is low. Study of the genetic structure, distribution pattern, and historical dynamics of *I. yunguiensis* in all areas of its distribution is of guiding significance for its rational and effective protection.

**Methods**. Expressed sequence tag-simple sequence repeat (EST-SSR) markers were used to study the genetic diversity and structure of *I. yunguiensis*, and noncoding chloroplast DNA (cpDNA) sequences were used to study the pedigree, population dynamics history, and glacial shelter of *I. yunguiensis*. A maximum entropy model was used to predict the past, present, and future distribution patterns of *I. yunguiensis*.

**Results**. Analysis with EST-SSR markers revealed that *I. yunguiensis* showed high genetic diversity and that genetic variation was significantly higher within populations than between populations. Based on cpDNA data, it was concluded that there was no significant geographic pedigree in the whole area of *I. yunguiensis* distribution (NST = 0.344 > GST = 0.183, $p > 0.05$); 21 haplotypes were detected using DnaSP v5. Neutral test and LAMARC simulation showed that *I. yunguiensis* has experienced rapid expansion in recent years. The maximum entropy model predicted that the potential distribution area of *I. yunguiensis* in the last glacial maximum period has increased significantly compared with the present distribution area, but the future distribution area did not show substantial changes.

## INTRODUCTION

Phylogeography, first proposed by *Avise et al. (1987)*, traces the evolutionary history of populations and explains the historical distribution of existing biota and historical causes of population differentiation. The Yunnan–Guizhou Plateau is one of the tropics of biodiversity in the world (*Myers et al., 2000*). Located in the southwest of China, the plateau has an area of $3 \times 10^5$ km$^2$ and an elevation of 1,000–3,000 m, with a higher elevation in the west and lower elevation in the east. It is divided into the Yunnan and Guizhou Plateaus, with the Wumeng Mountains as its boundary. The Yunnan Plateau

Corresponding author
Guangqian Gou, gqgou@gzu.edu.cn,
ggqian106@163.com

is located in the higher elevation area in the west, and the Guizhou Plateau is located in the lower elevation area (*Cheng et al., 2001*). The Yunnan–Guizhou Plateau has played an important role in revealing the biological consequences of Cenozoic orogenic events. Since the Cenozoic, the Yunnan–Guizhou Plateau has been continuously uplifted, which has resulted in unique geomorphological structures and complex land conditions (*Wu et al., 2008*). In recent years, the phylogeography of some plants in the Yunnan–Guizhou Plateau have been revealed (*Li et al., 2012*; *Wang, Wang & Su, 2014*). Therefore, it is necessary to study the pedigree history and distribution pattern of the endemic plant *Isoetes yunguiensis* Q. F. Wang & W. C. Taylor (*Wang et al., 2002*) in the Yunnan–Guizhou Plateau.

*Isoetes* L., the only genus of Isoetaceae, probably evolved from *Annalepis* (*Meng, 1998*). The evolutionary history of *Isoetes* almost spans the entire evolutionary history of vascular plants, and it is the sole extant representative of plant groups that evolved with simplified plant bodies, there by lending considerable convenience in studies regarding the origins and evolution of pteridophytes (*Zhang & Taylor, 2013*). Six species of *Isoetes* are found in China (*Li, 2017*); among these, *Isoetes orientalis* H. Liu & Q. F. Wang and *Isoetes shangrilaensis* Xiang Li, Yuqian Huang, Xiaokang Dai & Xing Liu were recently described (*Liu, Wang & Taylor, 2005*; *Li et al., 2019a*; *Li et al., 2019b*) and were shown to exhibit clear stepped distribution patterns.

*I. yunguiensis* Q. F. Wang & W. C. Taylor is a perennial quillwort endemic to China (http://www.iplant.cn/rep/), and it is the first plant to receive national level I protection (*Yu, 1999*). In recent years, the species populations have substantially declined, and the plant has recently been listed as a critically endangered (CR) plant in China by *Dong et al. (2017)*. *I. yunguiensis* was once sporadically distributed in the northern suburbs of Kunming (Yunnan Province) as well as in Xindan Tiansheng Bridge and Pingba (Guizhou Province). With climatic and environmental changes, wild *I. yunguiensis* populations are gradually declining, and several historical populations in Yunnan have already disappeared. Therefore, protection of the existing populations is paramount. *I. yunguiensis* is quite similar in appearance to *Isoetes japonica* A. Braun and was considered as *I. japonica* for a long period. Renchang Qin, for the first time, distinguished *I. yunguiensis* from *I. japonica* and described it as *I. yunkweiensis* Ching. Owing to the lack of complete information, this finding was never published (*Liu et al., 2002*). Subsequently, *Zhang (2001)* re-described this species as *I. chingiana* to honor Qin's research; however, this name too was never published (*Liu et al., 2002*). The species was finally described and published as the Chinese endemic *I. yunguiensis* (*Wang et al., 2002*). In a summary of previous studies, *Wang et al. (2002)* reported clear differences between the two species. In particular, *I. yunguiensis* megaspores are protuberant and numerous compared with *I. japonica* megaspores, and both species have a distinct number of chromosomes ($2n = 22$ in *I. yunguiensis* vs. $2n = 66$, 67, 77, 88, or 89 in *I. japonica*). Recent studies on *I. yunguiensis* have mainly focused on its morphology (*Zhao, Yan & Liu, 2015*), palynology (*Liu et al., 2013*), and molecular genetics (*Ma et al., 2018*; *Dong et al., 2018*). However, to the best of our knowledge, no studies on its evolution and biogeography have been published. Phylogeographic and genetic diversity studies on *I. yunguiensis* provide the basic data necessary for determining relationships across the evolutionary lineages of alpine plants during the Quaternary glacial period. Such

relationships may provide a basis for the protection of endangered plants. Here, expressed sequence tag-simple sequence repeat (EST-SSR) markers and chloroplast DNA (cpDNA) were used to determine the genetic diversity and structure of *I. yunguiensis* populations. The distribution and evolutionary patterns of genetically distinct *I. yunguiensis* groups were revealed, and the causes of the present distribution patterns of *I. yunguiensis* were discussed, which may provide scientific evidence for the protection of *I. yunguiensis*.

## MATERIALS & METHODS

### Materials

After obtaining approval from the Guizhou Province Wild Flora and Fauna Management Station, a total of 167 samples were collected from 14 populations in Yunnan and Guizhou Provinces from October 2018 to January 2019 (Table 1). Briefly, 8–15 samples that were distributed 10 m apart were randomly selected from each site (random sampling of minimal population). Healthy young leaves were collected in paper bags, dried immediately with silica, and stored at −20 °C for further use. Voucher specimens were stored in the Guizhou Agricultural College Plant Protection Department Herbarium (GACP). Figure 1 presents a map of sampling locations and geographical distribution of haplotypes.

### DNA extraction, polymerase chain reaction (PCR) amplification, and sequencing

DNA was extracted from dried leaves using the TIANGEN Plant Genomic DNA Extraction Kit [DP305, TIANGEN BIOTECH (BEIJING) CO., LTD.]. PCR analyses were performed using the EST-SSR primers SJSSR5, SJSSR12, SJSSR14, SJSSR45, and SJSSR53 and the cpDNA primers trnS-trnG, psbC-trnS, and psbD-trnT (Table 2). PCR reactions were conducted in 25-μL mixtures comprising 12.5 μL 2× T5 Super PCR Mix (Beijing TsingKe Biotech Co., Ltd.), 40 ng DNA template, and 1 μL each of (10 pmol/μL) forward and reverse primers, topped-up to final volume with double-distilled water. EST-SSR PCR was performed with an initial denaturation step at 94 °C for 5 min, followed by 28 cycles of denaturation at 94 °C for 30s, annealing at respective temperatures for 30 s, extension at 72 °C for 1 min, and final extension at 72 °C for 30 min. Reactions with cpDNA primers were performed with an initial denaturation step at 94 ° C for 5 min, followed by 35 cycles of denaturation at 94 °C for 30 s, annealing at respective temperatures for 30 s, extension at 72 °C for 1 min, and final extension at 72 °C for 10 min. EST-SSR PCR products were visualized as clear bands using 1.5% agarose gel electrophoresis and detected using capillary electrophoresis with ABI 3730 (Thermo Fisher Scientific). GeneMaker 2.2 was used to analyze the original peak patterns and to determine the sizes of EST-SSR marker fragments. Similarly, cpDNA PCR products were visualized as clear bands using 1.5% agarose gel electrophoresis, purified via electrophoretic cutting and using magnetic beads, and sequenced using ABI 3730. All sequences have been deposited in GenBank under the accession numbers MN463102–MN463268 for psbC-trnS, MN463269–MN463435 for psbD-trnT, and MN463436–MN463602 for trnG-trnS.

**Table 1  Details of sampling locations of the 14 Isoetes *yunguiensis* populations.** Coordinates and number of individuals sampled (Nind) are shown for each population.

| Population code | Population geographic location | Latitude | Longitude | Altitude (m) | Nind |
|---|---|---|---|---|---|
| GJS | Gengjia Mountain, Longli Forest Farm | 106°57′57″E | 26°29′45″N | 1276 | 8 |
| DZ | Dazhu Village, Longli County | 106°56′50″E | 26°31′29″N | 1281 | 11 |
| XB | Xinbai, Huishui County | 106°54′20″E | 26°04′40″N | 1289 | 15 |
| LS | Hongxing Village, Longli County | 106°50′10″E | 26°18′31″N | 1445 | 15 |
| GP | Huaxi, Guiyang City | 106°49′41″E | 26°16′43″N | 1461 | 14 |
| BSH | Baishuihe, Huishui County | 106°56′13″E | 26°04′51″N | 1434 | 10 |
| CX | Zixi Mountain, Chuxiong City | 101°24′14″E | 25°00′55″N | 2447 | 12 |
| DJC | Luyuan Resort, Longli County | 106°54′25″E | 26°21′48″N | 1603 | 11 |
| DPJ | Dapingjing Wetland Park, Nayong County | 105°27′41″E | 26°40′10″N | 2024 | 12 |
| XLC | Xinglongchang, Xingren County | 105°05′03″E | 25°24′35″N | 1540 | 15 |
| PB | Pingba, Anshun City | 106°17′07″E | 26°25′29″N | 1316 | 14 |
| SCZ | Shuichanzhan, Nayong County | 105°23′17″E | 26°44′05″N | 1677 | 8 |
| SG | SG, Longli Forest Farm | 106°56′15″E | 26°27′25″N | 1229 | 12 |
| HFH | Hongfenghu, Qingzhen City | 106°24′13″E | 26°30′01″N | 1250 | 10 |

# DATA ANALYSIS

## DNA analysis

Observed allele number (Na), effective allele number (Ne), observed heterozygosity (Ho), and expected heterozygosity (He) were calculated, and principal coordinate analyses (PCoA) and Mantel tests (*Mantel, 1967*) were performed using GenAlEx 6.5 (*Peakall & Smouse, 2012*). Polymorphism information content (PIC) was calculated using CERVUS 3.0 (*Kalinowski, Taper & Marshall, 2007*). Nei's genetic distances were calculated using POPGene 32 (*Yeh et al., 2000*), and Bayesian clustering was performed using STRUCTURE 2.3.3 (*Pritchard, Wen & Falush, 2009*). The original peak images of noncoding cpDNA regions were manually evaluated using Chromas 2.6, whereas the ClustalW program in the MEGA 7.0 software was used to align and truncate the three gene fragments and examine whether poly structure existed or were deleted. Concatenate Sequence program in PhyloSuite v1.1.15 (*Zhang et al., 2018*) was used to connect the three gene fragments into one fragment. Insertion–deletions are considered mutation sites (*Chen et al., 2008*). DnaSP v5 (*Librado & Rozas, 2009*) was used to determine variation sites,

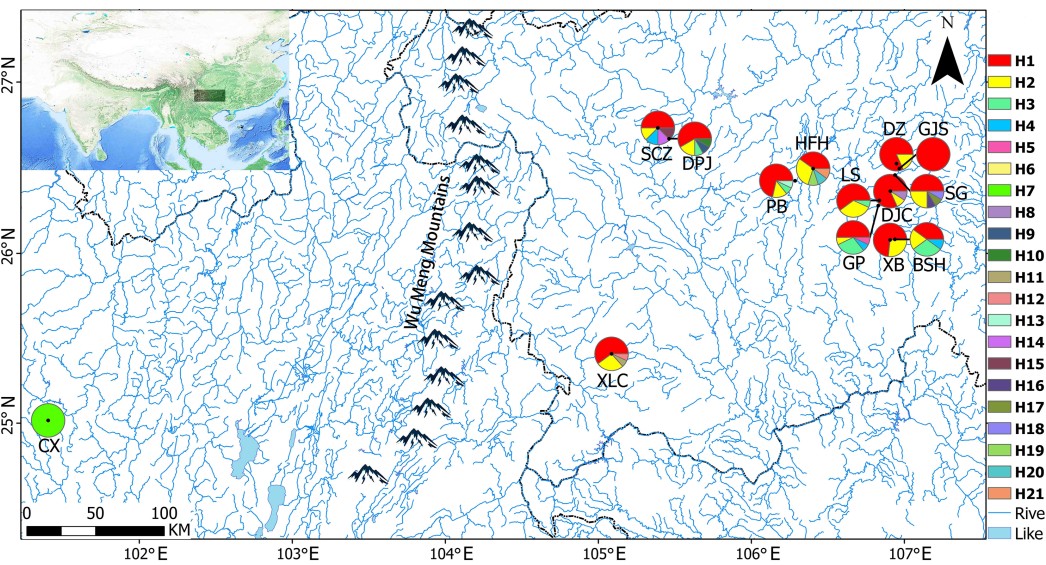

**Figure 1 Sampling locations and geographical distributions of haplotypes.** The pie charts show the proportions of different haplotypes in each population.

**Table 2 Information of primer pairs used to amplify by PCR.**

| Primer | Sequence (5′-3′) | Tm (°C) | Reference |
|---|---|---|---|
| SJSSR5 | **F:** CACACCCAACATACATACGCA, **R:** ATGGGGTGAAACAACAGGAG | 53 | *Gichira et al. (2016)* |
| SJSSR12 | **F:** GTGGTTGATTTGGGGTCATC, **R:** CCCTCTTTGCCAACAGTGAT | 53 | *Gichira et al. (2016)* |
| SJSSR14 | **F:** GGCCAGAGAACAGGAGAAAG, **R:** CCAAGTGGAAATTATGTCGCT | 55 | *Gichira et al. (2016)* |
| SJSSR45 | **F:** AAGGCCAACACAAAAACTGG, **R:** CGCCCACTAATCAGGACACT | 52 | *Gichira et al. (2016)* |
| SJSSR53 | **F:** AGGGATTGCTAGCGCTGTTA, **R:** GGCAAAACAAAAGCATCCAT | 54 | *Gichira et al. (2016)* |
| psbC-trnS | **F:** TGAACCTGTTCTTTCCATGA, **R:** GAACTATCGAGGGTTCGAAT | 52 | *Nishizawa & Watano (2000)* |
| psbD-trnT | **F:** CTCCGTARCCAGTCATCCATA, **R:** CCCTTTTAACTCAGTGGTAG | 55 | *Shaw et al. (2007)* |
| trnS-trnG | **F:** AGATAGGGATTCGAACCCTCGGT, **R:** GTAGCGGGAATCGAACCCGCATC | 52 | *Shaw et al. (2007)* |

haplotype diversity (Hd), and nucleotide diversity ($\pi$) of cpDNA sequences, and neutral tests and mismatch distribution analyses of cpDNA fragments were performed at the species level. LAMARC 2.1.8 was used to further verify the group expansion events for *I. yunguiensis*, and Permut 2.0 (*Pons & Petit, 1996*) was used to calculate total genetic diversity (Ht) and intrapopulation mean genetic diversity (Hs). To construct haplotype phylogenetic trees, analysis of molecular variance (AMOVA) was performed using Arlequin 3.5 (*Excoffier*

*& Lischer, 2010*) and IQ-TREE (*Nguyen et al., 2014*). A median-joining network map was constructed using Network 5.0 (*Bandelt, Forster & Röhl, 1999*).

## Species distribution prediction

MaxEnt 3.4 (*Phillips, Anderson & Schapire, 2006*) was used to predict the distribution ranges of *I. yunguiensis* during the last interglacial (LIG) period and last glacial maximum (LGM) as well as in the present and future. Coordinates of *I. yunguiensis* populations were determined using field records from the China Digital Herbarium (http://www.cvh.ac.cn/), as described previously (*Pang et al., 2003*; *Yuan et al., 2012*; *Li et al., 2015*). Maps were acquired from the National Catalogue Service for Geographic Information (http://www.webmap.cn). Climatic data were downloaded from the World Climate Data website (http://www.worldclim.org; resolution choice, 30 s). Correlations among 19 bioclimatic variables were calculated with the Raster Correlations and Summary Statistics tool in the SDMtoolbox. The environmental and extreme climatic factors with a correlation coefficient of $|r| \leq 0.8$ were selected. Finally, eight environmental variables were selected as the prediction variables of the distribution of *I. yunguiensis*: average daily range (bio 2), temperature seasonality (bio 4), maximum temperature of the warmest month (bio 5), average temperature of the wettest quarter (bio 8), average temperature of the coldest quarter (bio 11), precipitation of the wettest month (bio 13), precipitation of the driest month (bio 14), and precipitation of the warmest quarter (bio 18). Briefly, environmental data and spatial coordinates were imported into MaxEnt 3.4, parameters were set to default values, and distribution ranges during various periods were predicted. Subsequently, ArcGis 10.2 was used to generate ASCII raster layers and simulated distribution maps.

# RESULTS

## Genetic diversity of EST-SSR loci

The genetic diversity of EST-SSR loci in *I. yunguiensis* populations is presented in Table 3. Mean Na was 2.843, ranging from 1.929 (SJSSR53) to 4.000 (SJSSR12), mean Ne was 2.138, ranging from 1.521 (SJSSR53) to 2.981 (SJSSR12), and mean Shannon information index (I) was 0.782, ranging from 0.417 (SJSSR53) to 1.175 (SJSSR12). Ho ranged from 0 (SJSSR14) to 0.960 (SJSSR5), and He ranged from 0.273 (SJSSR53) to 0.642 (SJSSR12). Mean Ho and He were 0.475 and 0.463, respectively. PIC was high at a mean of 0.516, ranging from 0.313 (SJSSR53) to 0.641 (SJSSR12). The genetic diversity was the highest for SJSSR12 loci and the lowest for SJSSR53 loci.

## Genetic diversity and structure of populations

The genetic diversity of different populations is summarized in Table 4. Mean Na was 2.843, ranging from 2.400 (LS) to 3.400 (CX and XB), mean Ne was 2.138, ranging from 1.777 (LS) to 2.675 (CX), and mean I was 0.782, ranging from 0.595 (LS) to 1.016 (CX). Ho and He ranged from 0.251 (XB) to 0.673 (DJC) and from 0.335 (XLC) to 0.591 (CX), with mean values of 0.475 and 0.463, respectively. The genetic diversity was high in BSH, CX, DJC, DPJ, GJS, SG, and PB and low in LS, GP, and XB.

AMOVA using EST-SSR markers showed that 22.25% of the total genetic variation was present between *I. yunguiensis* populations and up to 77.75% was present within

**Table 3  Polymorphism analysis of the EST-SSR primers.**

| Loci | Na (mean ± SD) | Ne (mean ± SD) | I (mean ± SD) | Ho (mean ± SD) | He (mean ± SD) | PIC |
|---|---|---|---|---|---|---|
| SJSSR5 | 2.929 ± 0.917 | 2.361 ± 0.437 | 0.899 ± 0.211 | 0.960 ± 0.042 | 0.565 ± 0.068 | 0.552 |
| SJSSR12 | 4.000 ± 0.555 | 2.981 ± 0.678 | 1.175 ± 0.228 | 0.828 ± 0.321 | 0.642 ± 0.113 | 0.641 |
| SJSSR14 | 2.857 ± 0.363 | 2.191 ± 0.467 | 0.869 ± 0.197 | 0 | 0.520 ± 0.122 | 0.553 |
| SJSSR45 | 2.500 ± 1.225 | 1.637 ± 0.576 | 0.550 ± 0.414 | 0.118 ± 0.158 | 0.316 ± 0.234 | 0.519 |
| SJSSR53 | 1.929 ± 0.616 | 1.521 ± 0.510 | 0.417 ± 0.319 | 0.468 ± 0.446 | 0.273 ± 0.228 | 0.313 |
| Mean | 2.843 ± 0.461 | 2.138 ± 0.334 | 0.782 ± 0.174 | 0.475 ± 0.203 | 0.463 ± 0.097 | 0.516 |

Notes.

Na, Number of alleles observed; Ne, Effective number of alleles; I, Shannons information index; Ho, Observed heterozygosity; He, Expected heterozygosity; PIC, Polymorphic information content; SD, standard deviation.

**Table 4  Genetic diversity of 14 Isoetes yunguiensis populations in analyses using EST-SSR markers.**

| Population | Na (mean ± SD) | Ne (mean ± SD) | I (mean ± SD) | Ho (mean ± SD) | He (mean ± SD) |
|---|---|---|---|---|---|
| GJS | 2.600 ± 1.140 | 2.116 ± 0.783 | 0.746 ± 0.464 | 0.600 ± 0.548 | 0.455 ± 0.264 |
| DZ | 2.800 ± 1.095 | 2.245 ± 0.943 | 0.803 ± 0.497 | 0.400 ± 0.509 | 0.466 ± 0.282 |
| XB | 3.400 ± 0.548 | 1.843 ± 0.610 | 0.749 ± 0.310 | 0.251 ± 0.380 | 0.407 ± 0.195 |
| LS | 2.400 ± 1.140 | 1.777 ± 0.685 | 0.595 ± 0.432 | 0.386 ± 0.529 | 0.362 ± 0.254 |
| GP | 2.600 ± 1.673 | 2.018 ± 1.025 | 0.645 ± 0.619 | 0.400 ± 0.548 | 0.370 ± 0.343 |
| BSH | 3.200 ± 1.304 | 2.407 ± 0.554 | 0.956 ± 0.275 | 0.578 ± 0.485 | 0.569 ± 0.087 |
| CX | 3.400 ± 1.342 | 2.675 ± 0.926 | 1.016 ± 0.352 | 0.667 ± 0.445 | 0.591 ± 0.127 |
| DJC | 3.000 ± 1.000 | 2.314 ± 0.388 | 0.909 ± 0.218 | 0.673 ± 0.466 | 0.559 ± 0.069 |
| DPJ | 2.800 ± 0.837 | 2.546 ± 0.973 | 0.913 ± 0.376 | 0.503 ± 0.432 | 0.550 ± 0.197 |
| XLC | 2.800 ± 1.304 | 1.818 ± 0.882 | 0.611 ± 0.537 | 0.333 ± 0.437 | 0.335 ± 0.303 |
| PB | 3.000 ± 0 | 2.151 ± 0.546 | 0.839 ± 0.223 | 0.443 ± 0.456 | 0.504 ± 0.155 |
| SCZ | 2.600 ± 0.894 | 2.013 ± 0.770 | 0.729 ± 0.381 | 0.425 ± 0.527 | 0.442 ± 0.213 |
| SG | 2.600 ± 0.894 | 1.960 ± 0.517 | 0.739 ± 0.251 | 0.567 ± 0.518 | 0.463 ± 0.133 |
| HFH | 2.600 ± 1.140 | 2.050 ± 0.918 | 0.697 ± 0.512 | 0.424 ± 0.490 | 0.410 ± 0.296 |
| Mean | 2.843 ± 0.275 | 2.138 ± 0.200 | 0.782 ± 0.104 | 0.475 ± 0.121 | 0.463 ± 0.058 |

Notes.

Na, Number of alleles observed; Ne, Effective number of alleles; I, Shannons information index; Ho, Observed heterozygosity; He, Expected heterozygosity; SD, standard deviation.

**Table 5  AMOVA of 14 Isoetes *yunguiensis* populations using EST-SSR markers.**

| Source of variation | d.f. | Sum of squares | Variance components | Percentage variation |
|---|---|---|---|---|
| Among populations | 13 | 89.536 | 0.25247 Va | 22.25 |
| Within populations | 320 | 282.234 | 0.88198 Vb | 77.75 |
| Total | 333 | 371.769 | 1.134 | Fst = 0.223[*] |

Notes.

d.f., degrees of freedom; Fst, degree of population differentiation.

[*]$p < 0.001$.

populations (Table 5). Mean interpopulation gene flow rate (Nm) was 1.534, which is considered high, indicating frequent gene exchange between populations. These observations are consistent with the lower genetic variation observed between populations than within populations.

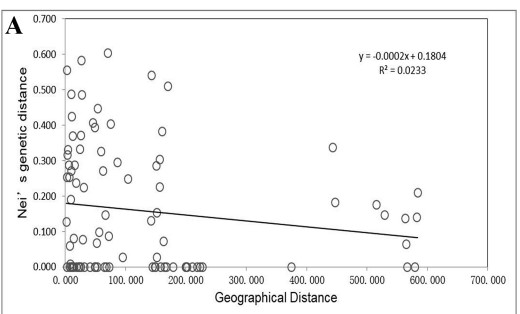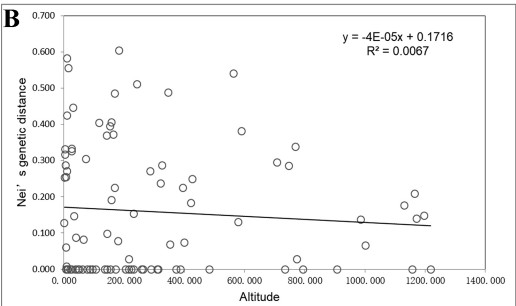

**Figure 2  Mantel tests of genetic distances and geographic distances/altitudes based on EST-SSR markers.** (A) Mantel tests of genetic distances and geographic distances based on EST-SSR markers; (B) Mantel tests of genetic distances and altitudes based on EST-SSR markers.

Mantel tests showed no significant associations of Nei's genetic distances with geographical distances ($p = 0.242 > 0.05$; Fig. 2A) or altitude gradients ($p = 0.402 > 0.05$; Fig. 2B).

## Genetic relationships and cluster analyses

Unweighted pair group method with arithmetic mean (UPGMA) cluster analyses were performed based on Nei's genetic distances using MEGA 7.0, and 14 UPGMA phylogenetic trees of *I. yunguiensis* populations were constructed (Fig. 3). With a genetic distance of 0.2, the 14 populations were divided into 2 groups. The first group CX comprised only one unique population from the Yunnan Plateau, whereas the second group comprised the remaining 13 populations from the Guizhou Plateau. With a genetic distance of 0.1, the second group was subdivided into the groups II-a and II-b. The group II-a predominantly comprised populations from the west of Guiyang City, whereas the group II-b comprised populations from to the east of Guiyang City; however, there was no notable difference between both groups. PCoA results (Fig. 4) were consistent with UPGMA results, in which the first and second components explained 51.50% and 18.76% of the variance, respectively.

STRUCTURE 2.3.3 was used to analyze the genetic structure of *I. yunguiensis* populations. At $K = 2$, $\Delta K$ reached the maximum value (Fig. 5), and *I. yunguiensis* individuals were divided into two groups (Fig. 6). The group I (blue; Fig. 6) mainly comprised individuals of the CX population, whereas the group II comprised individuals from the remaining populations (red; Fig. 6). Overall, 30 of the 167 individuals showed differences of different degrees, indicating a variable degree of gene exchange between populations.

## cpDNA sequence characteristics and haplotype analyses

After aligning and connecting the three cpDNA sequences from *I. yunguiensis*, the total length was 1,657 bp. Moreover, 36 mutation sites were detected, among which 11 were detected by trnS-trnG, including 8 single-base mutations and 3 insertion–deletion sites. Only two mutation sites were detected by psbC-trnS, and both were single-base mutations.

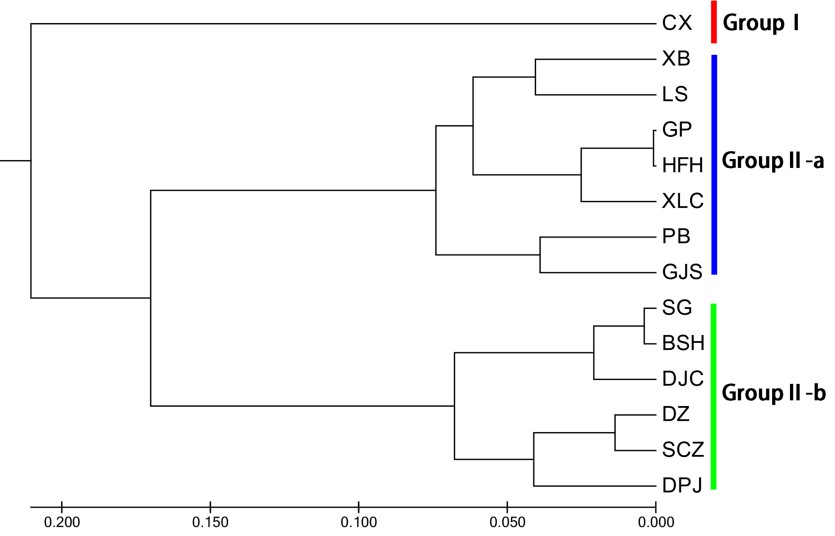

**Figure 3** UPGMA cluster analysis of 14 *Isoetes yunguiensis* populations based on EST-SSR markers.

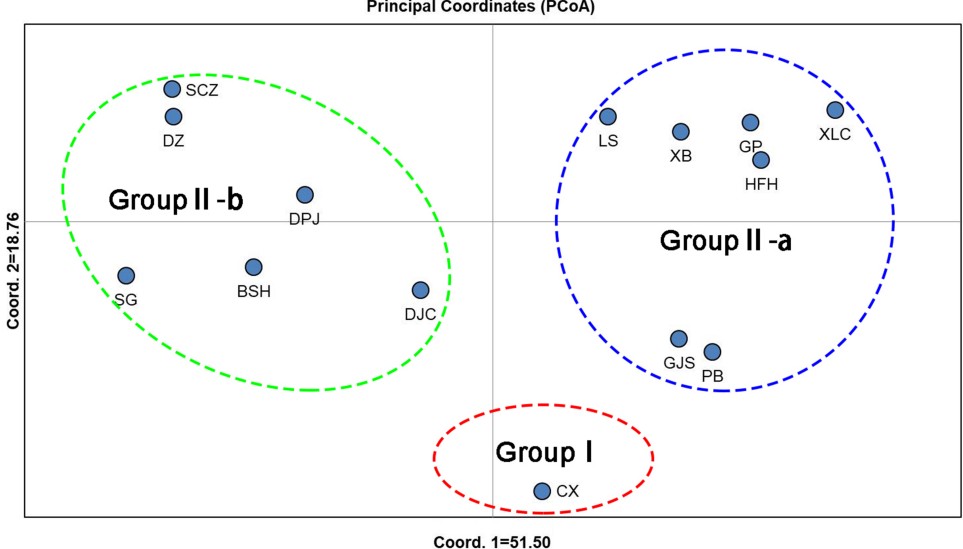

**Figure 4** PCoA analysis of *Isoetes yunguiensis* populations based on EST-SSR markers.

Overall, 23 mutation sites were detected by spsbD-trnT, including 20 single-base mutations and 3 insertion–deletions.

A total of 21 haplotypes were detected using DnaSP v5, and haplotype H1 was the most widely distributed haplotype. With the exception of the CX population, in all the remaining 13 populations, haplotype H1 was the most common, followed by haplotype H2. Haplotype H3 was distributed in GP, BSH, DPJ, LS, and PB populations, haplotype H4 was distributed in GP and SCZ populations, and the unique haplotypes H5–H21 were

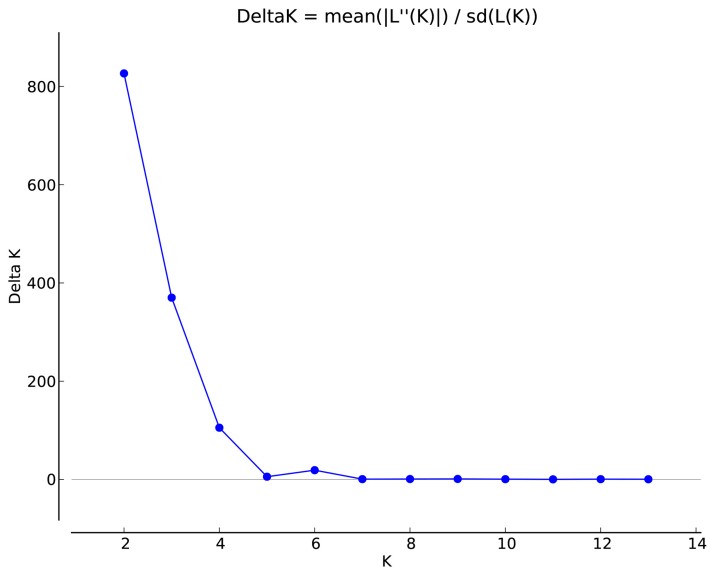

**Figure 5   Association between ΔK and K in the STRUCTURE analyses.**

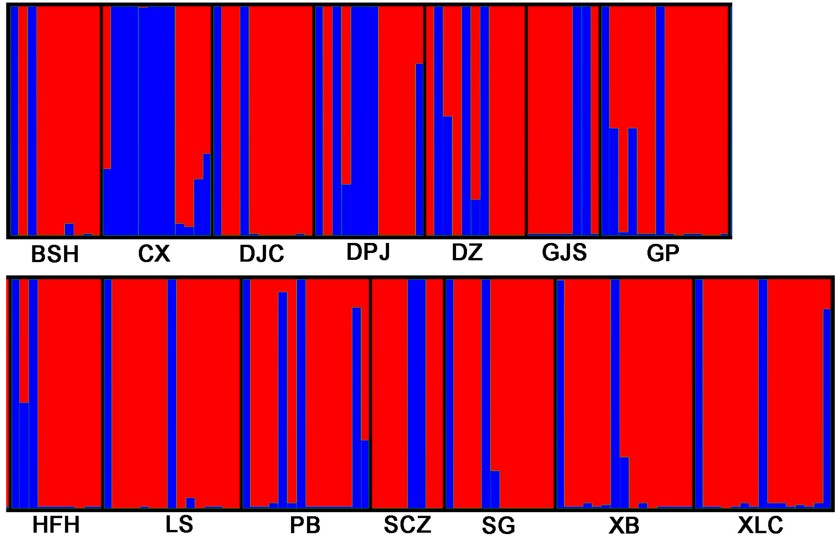

**Figure 6   STRUCTURE analyses of 14 *Isoetes yunguiensis* populations based on EST-SSR markers.**

distributed in only a single population, and only 1 CX population carrying the unique haplotype H7. Genetic diversity, nucleotide diversity, and haplotype compositions and frequencies of the sampled populations are shown in Table 6.

## Population genetic structure based on cpDNA sequences

The total genetic diversity (Ht) was 0.631, mean genetic diversity (Hs) was 0.515, and the genetic differentiation coefficients between (GST) and within (NST) populations were 0.183 and 0.344, respectively. The haplotype variation structure of the 14 *I. yunguiensis*

**Table 6  Haploidy and sequence characteristics of chloroplast DNA.**

| Population code | Haplotype (number) | Hd | π |
|---|---|---|---|
| GJS | H1 (8) | 0.00000 | 0.00000 |
| DZ | H1 (9), H2 (2) | 0.32727 | 0.00040 |
| XB | H1 (11), H2 (4) | 0.41905 | 0.00051 |
| LS | H1 (9), H2 (5), H3 (1) | 0.56190 | 0.00060 |
| GP | H1 (7), H2 (1), H3 (4), H4 (1), H5 (1) | 0.70330 | 0.00017 |
| BSH | H1 (4), H2 (2), H3 (3), H6 (1) | 0.77778 | 0.00012 |
| CX | H7 (12) | 0 | 0 |
| DJC | H1 (9), H2 (1), H8 (1) | 0.34545 | 0.00011 |
| DPJ | H1 (7), H2 (2), H3 (1), H9 (1), H10 (1) | 0.66667 | 0.00101 |
| XLC | H1 (9), H2 (4), H11 (1), H12 (1) | 0.60000 | 0.00048 |
| PB | H1 (10), H2(2), H3 (1), H13 (1) | 0.49451 | 0.00046 |
| SCZ | H1 (4), H2 (1), H4 (1), H14 (1), H15 (1) | 0.78571 | 0.00056 |
| SG | H1 (6), H2 (3), H16 (1), H17 (1), H18 (1) | 0.72727 | 0.00071 |
| HFH | H1 (4), H2 (3), H19 (1), H20 (1), H21 (1) | 0.80000 | 0.00034 |

**Notes.**
Hd, Haplotype diversity; $\pi$, Nucleotide diversity.

**Table 7  AMOVA of 14 Isoetes yunguiensis populations using cpDNA markers.**

| Source of variation | d.f. | Sum of squares | Variance components | Percentage variation |
|---|---|---|---|---|
| Between populations | 13 | 53.826 | 0.299 Va | 34.00 |
| Within populations | 153 | 88.875 | 0.581 Vb | 66.00 |
| Total | 166 | 142.701 | 0.880 | Fst = 0.34[*] |

**Notes.**
d.f., degrees of freedom; Fst, degree of population differentiation.
[*]$p < 0.001$.

populations was examined using 1000 bootstrap replicates, which revealed a tendency of NST > GST, although the difference was not significant ($p > 0.05$); therefore, *I. yunguiensis* exhibited no distinct phylogenetic structure in the studied areas. Consistently, AMOVA (Table 7) showed that the genetic variation between populations was 34.00% and that within populations was 66.00% (FST = 0.34, $p < 0.001$, 1000 bootstrap replicates). The genetic variation in *I. yunguiensis* was predominately higher within populations. Mean Nm was 2.18, which indicated substantial gene flow between *I. yunguiensis* populations.

## Population history dynamics based on cpDNA sequences

DnaSP v5 was used to examine the combined cpDNA sequence of all the individuals in the 14 sampled *I. yunguiensis* populations. Although the mismatch distribution was bimodal (Fig. 7), Tajima's D = −2.55856 ($p < 0.001$), Fu & Li's D = −8.03665 ($p < 0.02$), and Fu & Li's F = −6.94682 ($p < 0.02$) were significantly negative. The population growth index obtained by LAMARC simulation was 921.590 (G > 200). Taken together, these results indicated that the *I. yunguiensis* populations have experienced expansion events.
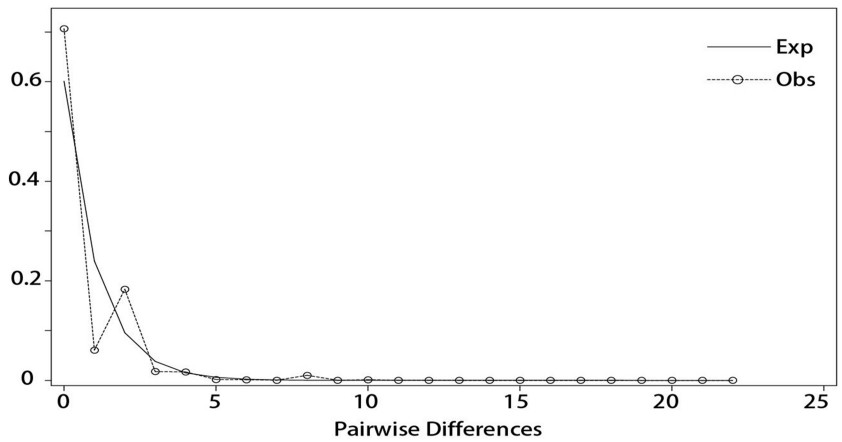

**Figure 7** **Mismatch distribution analysis among different inferred biogeographical groups.**

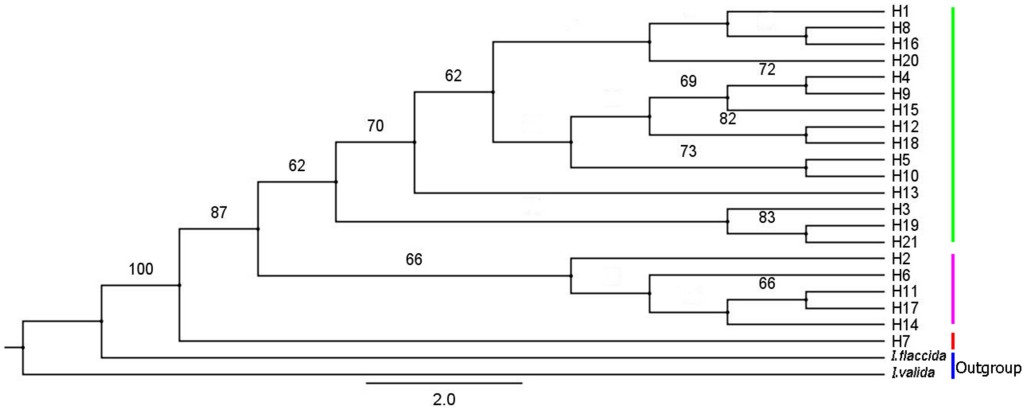

**Figure 8** **A phylogenetic tree of *Isoetes yunguiensis* haplotype constructed using IQ-TREE based on cpDNA.** The number above the node represents the bootstrap support rate (showing a value > 60).

## Phylogenetic relationships based on cpDNA haplotypes

IQ-TREE was used to construct a phylogenetic tree of *I. yunguiensis* haplotypes (Fig. 8). The model with the best HKY+F value was selected. *Isoetes flaccida* and *I. vallda* were used as outgroups. The 21 haplotypes were divided into two branches: the first branch included H1–H6 and H8–H21 and the second branch included H7 alone.

A mediation-link (median-joining) network diagram and phylogenetic tree based on the maximum likelihood method were constructed using Network 5.0 (Fig. 9). Because H1 and H2 appeared in the middle of the Network diagram, these were speculated to be relatively old haplotypes; moreover, there were few other haplotypes as well as variation coefficients were small, suggesting that recent haplotypes were derived from H1 and H2.

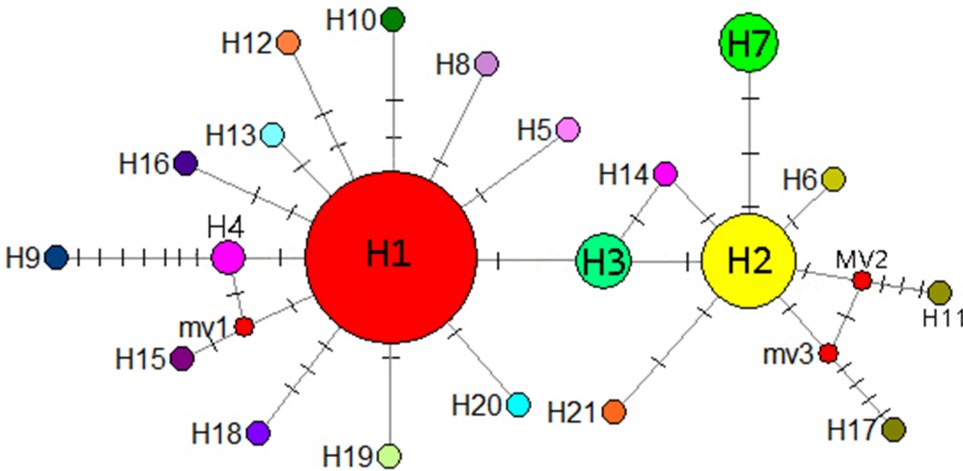

**Figure 9  Median-joining network reflecting haplotype relationships.** Small red circles represent potential intermediate haplotypes, The sizes of the circles are proportional to individual number of haplotype. The solid line indicates a base mutation step.

## Niche modeling

MaxEnt 3.4 was used to predict distribution ranges during the LIG period (Fig. 10A) and LGM (Fig. 10B) as well as in the present (Fig. 9C) and future (Fig. 10D). In all models, the areas under the curve were >0.947, indicating that the predicted distribution ranges of *I. yunguiensis* are located in the central Guizhou Plateau and northwestern and eastern Yunnan Plateau and that the suitable areas of distribution are mainly located in the central Guizhou Plateau. The future distribution pattern was similar to the present distribution pattern. The distribution range of *I. yunguiensis* during the LIG period was significantly reduced compared with the present distribution range, and the distribution ranges in the Yunnan–Guizhou Plateau were clearly separated. The distribution range during LGM was slightly expanded in the Yunnan Plateau and reduced in the Guizhou Plateau compared with the future distribution range.

## DISCUSSION

### Genetic diversity and structure

In the present study, *I. yunguiensis* showed high genetic diversity compared with other rare, endemic, and endangered plants, such as *Isoetes sinensis* Palmer (He = 0.118; *Kang, Ye & Huang, 2005*), *Isoetes hypsophila* Hand. Mazz. (He = 0.039; *Chen et al., 2010*), *Isoetes malinverniana* Ces. & De Not. (H = 0.1491 for ISSR data; H = 0.2289 for AFLP data; *Gentili et al., 2010*), *Ottelia acuminata* var. *jingxiensis* H. Q. Wang & S. C. Sun J. (He = 0.441; Li et al., 2019), *Dalbergia odorifera* T. Chen (He = 0.37; *Liu et al., 2019*), and *Brasenia schreberi* J. F. Gmelin (He = 0.256; *Li et al., 2018*). Most rare and endangered plants are seed plants that can achieve a higher gene flow via long-distance pollen and seed dispersal than ferns that rely solely on spores. *Ma et al. (2018)* investigated the genetic structure of *I. yunguiensis* using inter-SSR markers and showed inter- and intrapopulation genetic variations of

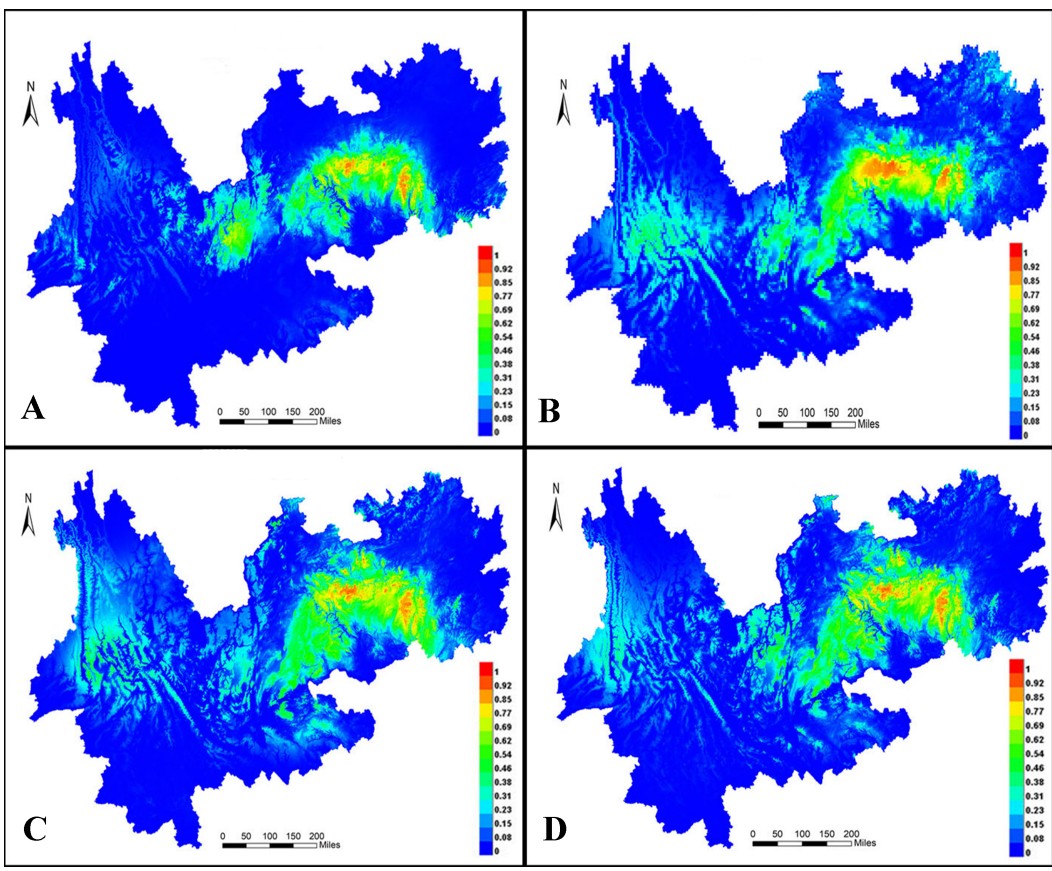

**Figure 10 Potential distribution modeling of *Isoetes yunguiensis* based on scenarios of the MIROC model.** (A) The last interglacial (LIG, ca. 130 Kya), (B) the last glacial maximum (LGM, ca. 22 Kya BP), (C) present day (average for 1970–2000), and (D) future in 2070 (average for 2061–2080).

31.99% and 68.01%, respectively. Similarly, *Dong et al. (2018)* used amplified fragment length polymorphism markers and showed inter- and intrapopulation variations of 40.12% and 59.88%, respectively. However, in these previous studies, samples were collected from Pingba and Hongfeng Lake alone, which limited the representativeness of the genetic variation of *I. yunguiensis*. The 14 sampled populations covered all areas of *I. yunguiensis* distribution, and the analyses of genetic structures and variations were conducted using EST-SSR and cpDNA sequences to reveal the genetic diversity of *I. yunguiensis*. Based on these markers, *I. yunguiensis* showed high genetic diversity. Moreover, AMOVA using these markers showed that the genetic diversity was higher within populations than between populations, which is consistent with the results of previous studies. Reportedly, in the areas of *I. yunguiensis* distribution, the observed haplotypes accounted for 80.95% of the total haplotypes, and this species often did not show clear biogeographical patterns (*Gao et al., 2012*; *Gao et al., 2016*). Accordingly, in the present study, the population genetic differentiation index was higher than the gene differentiation index, although the difference was not significant ($p > 0.05$), indicating that *I. yunguiensis* shows no distinct biogeographical patterns. *I. yunguiensis* is mainly dispersed through spores released in

flowing water (*Yang et al., 2011*), and it is distributed in the Yangtze and Pearl River water systems. In the present study, a high gene flow was observed between *I. yunguiensis* populations, which may explain the high genetic variation within populations (*Huang et al., 2017*).

Using STRUCTURE 2.3.3, the 14 *I. yunguiensis* populations were divided into two groups, and these results were consistent with those of UPGMA culture analysis, PCoA, and IQ-TREE based on Nei's genetic distances. The two groups are separated by the Wumeng Mountains, which follow the border between the Yunnan–Guizhou Plateaus. The mountains between these provinces have isolated these *I. yunguiensis* populations, thereby explaining the genetic distances between the two main haplotype groups. However, based on the STRUCTURE analysis, the genetic structure of *I. yunguiensis* is relatively complex. Further studies are required to elucidate specific reasons underlying the difference between the CX population from the Yunnan Plateau and the remaining 13 populations from the Guizhou Plateau. However, Mantel tests showed no significant associations of Nei's genetic distances with geographical distances or altitude gradients.

## Population structure and distribution ranges of *I. yunguiensis*

The genetic structures of extant plants can be traced back to the Quaternary glacial period. Based on this, Yu et al. (2013) have summarized the genetic structures of 36 species of alpine plants from the Qinghai–Tibet Plateau and surrounding areas. Some hardy plants did not migrate to lower altitudes during the glacial episodes; however, plants on the plateau surface experienced small-scale expansions of ranges following LGM. The genetic structures of species are often characterized by high variations or specific haplotypes, with population distributions in the highland mesa of one or more isolated areas. In addition, some species occupied narrow ranges during the glacial period, leading to small areas of expansion following LGM. These populations were characterized by high genetic diversity and unique haplotypes distributed evenly throughout the distribution range. The present study demonstrated that although *I. yunguiensis* shows high genetic variation, regional genetic variation that is not linked to longitude and latitude or elevation is present, presumably because during the Quaternary glacial period, the Yunnan–Guizhou Plateau remained free of ice. Therefore, the influence of glacial episodes on *I. yunguiensis* was only characterized by the influences of changes in climate and availability of regional water, which affect the habitat and dispersal of *I. yunguiensis*. Few ancient haplotypes may have randomly been fixed in the population during migration, and young haplotypes could have subsequently been fixed in the population because of genetic drift and the founder effect, thereby forming the current population structure of *I. yunguiensis*.

*Hewitt (2000)* identified a small population that migrated to form populations with greater genetic variation and richer haploid types, potentially reflecting the duration of isolation and accumulation of genetic variants. When diffusion distances increase from outer diffusion migration groups that are prone to genetic drift or the founder effect, haplotype polymorphism is gradually reduced. According to the coalescent theory, populations located in refuge show high genetic diversity and ancient haplotypes are often located in the center of the haplotype network maps. Therefore, the geographical locations

of populations with high genetic diversity and ancient haplotypes can be used as bases for identifying potential habitats. In the present study, H1 and H2 were located in the center of the haplotype network map and may thus be ancient haplotypes that are widely distributed in the Guizhou Plateau. The CX population from the Yunnan Plateau comprised the unique and stable haplotype H7 and is located far from the Guizhou Plateau. It is possible that *I. yunguiensis* occupied specific areas of the Guizhou and Yunnan Plateaus during the glacial period. Because the ancestral population from the Yunnan Plateau is extinct, only one population was noted in the Yunnan Plateau in the present study, and the specific location of the extinct population in this plateau remains unknown.

## Conservation strategies for *I. yunguiensis*

The present study demonstrated that the BSH, CX, DJC, DPJ, GJS, SG, and PB populations showed high genetic diversity. It is important to understand genetic diversity and population structure to establish scientific and effective protection measures. Therefore, to protect the endangered *I. yunguiensis* plants, a provincial nature reserve has been established; the DPJ population used in the present study was sampled from this reserve. However, it has been difficult to protect this population. The SCZ population in the present study was sampled from a farm where sewage is directly discharged into the habitat. The DJC population was sampled from the middle of Longli Forest Farm; seven populations were sampled from this farm, accounting for half of the surveyed populations. However, although these populations were not under the threat of sewerage exposure, limited attention and protection have led to a state of low survival and even extinction. Therefore, the establishment of a protected area dedicated to *I. yunguiensis* in Longli Forest Farm and training of staff and residents of the surrounding areas regarding protective measures for *I. yunguiensis* are warranted. In a previous study, populations with high genetic diversity were selected from similar environments for transplantation (*Zhang, 2016*). According to the principles of genetic diversity and conservation prioritization in special areas, the HFH populations, which showed the highest haplotype diversity (Hd = 0.8) should be granted the highest priority, followed by the BSH, CX, DJC, DPJ, GJS, SG, and PB populations. As an overall conservation strategy based on the molecular genetic evidence provided in the present study, the Guizhou and Yunnan Plateaus can be divided into two areas to protect populations with high genetic diversity as well as those with high and stable haplotype diversity. The CX populations from the Yunnan Plateau and the BSH, HFH, DJC, SG, DPJ, GJS, and PB populations from the Guizhou Plateau should be primarily protected. Both the Guizhou and Yunnan Plateaus should be protected, with particular focus on the CX, BSH, and HFH populations. Moreover, the water quality in *I. yunguiensis* habitats should be maintained to avoid contamination, and the growth of companion species should be controlled. Improved knowledge and awareness regarding the areas of *I. yunguiensis* distribution are paramount. To strengthen the conservation strategies for endangered species, further studies are required to identify threats. Moreover, considering that the number of wild *I. yunguiensis* individuals included in this survey was <10,000, the populations are declining further.

## CONCLUSIONS

In the present study, *I. yunguiensis* exhibited higher genetic diversity than other rare and endangered plants, and its genetic structure demonstrated is similar to that previously reported. Genetic variation was significantly higher within populations than between populations. Based on the data of cpDNA, although no distinct pedigree was detected in the entire area of *I. yunguiensis* distribution, there has been rapid expansion in recent years. The distribution pattern simulated using the MaxEnt model showed that the past and future distribution patterns of *I. yunguiensis* are not considerably different from its present distribution pattern. EST-SSR marker and cpDNA data support that the 14 populations examined in the present study can be divided into two groups: those in the Yunnan Plateau and those in the Guizhou Plateau. This indicated that there were at least two refuges for *I. yunguiensis* during the glacial period. The Wumeng Mountains between the two plateaus have a substantial influence on gene exchange in *I. yunguiensis*. Based on the high genetic diversity and ancient haplotypes of the populations associated with the refuges, it can be concluded that the CX population from the Yunnan Plateau and the BSH, HFH, DJC, SG, DPJ, GJS, and PB populations from the Guizhou Plateau should be primarily protected. These plateaus are ice-age refuges where these populations should be protected. In the present study, only one population was identified in the Yunnan Plateau. Further investigation is required to identify the new population and discuss the area of refuge. To date, the cause of *I. yunguiensis* endangerment remains unclear, and related research is urgently warranted.

## ACKNOWLEDGEMENTS

We thank Mrs. Ya Zhang (Nayong Forest Farm), Professor Chenghua Yang (Guizhou Provincial Academy of Forestry), Mr. Jirong Feng et al., (Longli Forest Farm), and Qi Wei for sampling. We also thank Professor Cai Zhao and Professor Guoxiong Hu for their guidance.

### Funding

The authors received no funding for this work.

### Competing Interests

The authors declare there are no competing interests.

### Author Contributions

- Tao Zheng conceived and designed the experiments, performed the experiments, analyzed the data, contributed reagents/materials/analysis tools, prepared figures and/or tables, authored or reviewed drafts of the paper, approved the final draft.
- Xuanze He performed the experiments, prepared figures and/or tables, approved the final draft.

- Honghuan Ye conceived and designed the experiments, prepared figures and/or tables, authored or reviewed drafts of the paper, approved the final draft.
- Wei Fu and Maimai Peng performed the experiments, contributed reagents/materials/-analysis tools, prepared figures and/or tables, approved the final draft.
- Guangqian Gou conceived and designed the experiments, authored or reviewed drafts of the paper, approved the final draft.

## Field Study Permissions

The following information was supplied relating to field study approvals (i.e., approving body and any reference numbers):

The Guizhou Province Wild Flora and Fauna Management Station approved the collection of Isoetes yunguiensis leaf samples.

## Data Availability

Raw data from the cpDNA sequences (MN463102–MN463602) were applied for data analyses and preparation of Figs. 7–9 and Tables 6 and 7.

All voucher specimens are preserved in The Guizhou Wildlife Conservation Station (GACP). Specimen numbers are GJS2019011702, DZ2019011703, XB2018120502, LS2018120504, GP2019120503, BSH2018120501, CX2019011501, DJC2019011801, DPJ2019010401, XLC2018102901, PB2019010301, SCZ2019010402, SG2019011701, HFH2019010302.

## Supplemental Information

Supplemental information for this article can be found online at http://dx.doi.org/10.7717/peerj.8270#supplemental-information.

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
