# Peer review of "Phylogeography of the rare and endangered lycophyte Isoetes yunguiensis"

_PeerJ, doi:10.7717/peerj.8270_

## Round 0.1 · original submission · Major Revisions

Dear Dr. Zheng,
thank you very much for your submission to PeerJ. Your manuscript which you submitted to the journal has been kindly reviewed by the reviewers. In general, the reviewers find the manuscript original, well written and interesting, so they find the manuscript worthy of publication. Nevertheless, they detected some weaknesses that you have absolutely to care of. Please answer careful and in detail to all comments of the reviewers.
Once again, thank you for submitting your manuscript to PeerJ and we look forward to receiving your revision.
Sincerely,
Gabriele Casazza

·

Basic reporting

The English is sufficient, but it can be improved.

Literature references are good enough, even though some additional suggestions have been made.

Structure, figures, tables are good.

Results are adequately discussed and logical conclusions are related to aims as set out in the Introduction.

Experimental design

This paper deals with an interesting topic, within the scope of the journal.

The aims are clearly stated and introduced, the results and facts presented clearly and sufficiently fully and separate from interpretations. There is an appropriate statistical analysis of data.

Validity of the findings

Findings are interesting with general implications.

Additional comments

Some general issues:
1) Manuscript has to be revised to make it more coherent: see what is written in the abstract ("genetic variation between populations was significantly higher than that within populations") and in the Results ("lower genetic variation between populations than within populations").
2) the term "fern" (in the title, introduction etc.) seems not correct (see attached file).
3) Manuscript has to be revised to make it more fluid for English;
4) Some additional references have been suggested to improve discussion (about dispersal, and comparison with the genetic structure of other Isoetes spesies).

Additional notes in the attached file.

Reviewer 2 ·

Basic reporting

Summary – This manuscript focus on phylogeography of a rare and endagered plants (Isoetes yunguiensis) in China. The purpose was to identify gene-flow and genetic structure and dynamic of this species and use these finding to identify population suitable for conservation. The manuscript need some revision but provide interesting results concerning genetic diversity and dispersal within the fern allies clade.
The writing is clear.
Introduction - The taxonomy of I. yunguiensis is well described and the context of the manuscript (in terms of gap of knowledge and usefulness of the finding) clear. Considering the focus of the paper (i.e. Phylogeography), phylogeographic or biogeographic information is missing in the introduction. I will encourage authors to improve this section with the addition of relevant informations concerning the actual knowledge for the studied region and for analysis of similar species in other geographical context. Finally, I will suggest to formulate more explicit hypotheses.
Matherial and Method – line 81 : authors cite a collection of water samples, but no additional details are provided in the text (samples were used for some analysis, stored for other use?). If these samples were used for pH analysis (reported at line 310) why only pH was analyzed considering that one of the conservation issue could be related by eutrophic state?
On the PCR description I would like to have a reference citing the primer composition and choice, in addition brand providing reagent is missing, as well as annealing temperature. No informations about PCR purification are provided.
I will suggest to authors to improve data analysis paragraph. To improve understanding can be subdivided in two small paragraph (species distirbution prediction and DNA analysis).
Statistical part need improvement (e.g., informations concerning how indels are treated in the analysis are missing (e.g., 5th state?), if gene were used concatenated or not, how alignment was performed, ...). Moste of theses informations occour within the Results section, but this setting make understanding of the metholody more difficult.
I will also expand the distirbution/range section: for example, adding details on the bioclimatic variables included, if checked and corrected for correlation and their biological relevance on predicting Isoetes range, which setting were used, and so on. I’m not familiar with MaxEnt software, but I would also be interested in knowing if a 1km resolution is suitable for predicting species occourence in case of plants with this size, and additional information concerning number of herbarium datapoints.
Discussion - I appreciate the final opening with a practical-conservation oriented discussion. Therfore I think is too much considering the phylogeographic focus of the paper and need to be reduced (it take half of the length of discussion).

Figures – Fig. 1 I will remove color from background to improve the readibility of the image. I will try to make bigger pie chart and/or to reduce colors of haplotypes (e.g., selecting 3-4 main colors for the principles groups, can be assigned the same used in Fig. 8) in order to make more clear the main pattern.
Fig. 8 in captation for self-explanation I will add the origin of data (cpDNA), I will specify what the number represent (bootstrap or probability) and that only values higher than 60 are shown. Outgroup instead of out group.
Fig. 9 as already marked, in the main text is described as a figure composed by several panels , but only the network is displayed.
General remark : I will add more details on the legends of graphs in order to make each one self explaining (e.g., origin of genetic material is often missing).

Tables – In tab. 1 on line n.6 (BSH) altitude is probably wrong (134m, when all the other sites are > 1200m). Longitude need uppercase.
In tab. 2 and tab. 3 only mean is reported, standard deviation should be added.
For Tab. 4 check also Fst (which is in % but in Tab. 6 is not).
Tab. 4 I will add an additional column for F statistiscs and associated p-value. Then for self-explanation I will add on note the explanation of Va, Vb, and F-stat; and the number of permutation used to obtain the associated p-value. The same for Tab. 6. In Tab. 6 I will also add the origin of genes. Finally check in the text : Tab. 6 is not cited (I assume that probably is a mistake in line n. 194).

Supplementary material – DNA file can be opened correctly, and numbers are correct in according to what reported in the text. Each DNA sequences is labelled using the same code used within the text. Unfortunately no additional metadata are added in order to make the supplementary material self-explaining (e.g., information related to the different genes sequeced, additional informations on origin of the DNA material). In addition data are provided only for chloroplast DNA and no raw data for EST-SSR are provided.
Finally no information within method section are provided on where data will be available (e.g., genbank repository, Supplementary material, ...).

Experimental design

Probably limited by costs and material quality. But the adding of additional population better covering the region can be an improvement for understanding haplotype connections and biogeographical pattern. Why author not used herbarium samples cited on lines 122-23?

Explanation of statistical analyses need improvement (reshaping MM section as suggested before). Methods used are appropriates.

Validity of the findings

Results provided support the findings of authors. I think the strong genetic homogeneity is interesting and surprising considering the deep relationship within water for spore dispersal .

---

## Round 0.2 · Minor Revisions

Dear Dr. Zhang,

The reviewers find the manuscript strongly improved. Nevertheless, a reviewer requests some changes. I demand you to improve the paragraph ”species distribution prediction” in M&M. In particular, you have to specify how you tested multicollinearity between environmental predictors and what variables you retained for the analysis. In the case you didn't test for correlation between predictors and you retained all bioclimatic variables, you have to clearly say why you excluded that correlation between variables and the use of a high number of variables may not lead to overfitting and collinearity issues in your study case.

Once again, thank you for submitting your manuscript to PeerJ and we look forward to receiving your revision.

Sincerely,
Gabriele Casazza

·

Basic reporting

The manuscript is now clearer (but see notes about introduction, in the attached file).

Experimental design

The experimental design is now better explained.

Validity of the findings

The findings are interesting also for other cases. Discussion has been improved (but see attached file about IUCN category).

Additional comments

The manuscript has been improved, following my previous suggestions, but I ask you to change something (in the introduction and in the conservation strategies).

Reviewer 2 ·

Basic reporting

The authors have made the requested changes and answered the doubts raised above.
The text, tables and images are currently clearer.

Experimental design

no comment

Validity of the findings

no comment

---

## Round 0.3 · Minor Revisions

Dear Dr. Zhang,

You performed all changes suggested by reviewers and so the article is very close to being accepted. Nevertheless, some English mistakes still persist. So, before acceptance the text needs to be thoroughly edited for English. You might ask to your colleagues to carefully check the text or ask for a professional service.

Some examples are:

“Approved by the Guizhou Province Wild Flora and Fauna Management Station” - This probably means that you have an authorization to sample but is not clear.
“warmest month maximum temperature” - maximum temperature of the warmest month
“declinedsubstantially“ - declined substantially
“Coordinates of I. yunguiensis distribution” - probably populations or occurrences instead of distribution is better

Sincerely,
Gabriele Casazza

---

## Round 0.4 · accepted · Accept

Dear Dr. Zhang, the English was strongly improved. So, I am very pleased to say that your paper "Phylogeography of the rare and endangered lycophyte Isoetes yunguiensis" is accepted for publication in the PeerJ. Congratulations!

Thank you for submitting your work to PeerJ.

Yours sincerely,
Gabriele Casazza